# A Comparative Study of Mesenchymal Stem Cell-Derived Extracellular Vesicles’ Local and Systemic Dose-Dependent Administration in Rat Spinal Cord Injury

**DOI:** 10.3390/biology11121853

**Published:** 2022-12-19

**Authors:** Alexander Kostennikov, Ilyas Kabdesh, Davran Sabirov, Anna Timofeeva, Alexander Rogozhin, Ilya Shulman, Albert Rizvanov, Yana Mukhamedshina

**Affiliations:** 1Center for Clinical Research for Precision and Regenerative Medicine, Institute of Fundamental Medicine and Biology, Kazan Federal University, 420008 Kazan, Russia; 2Department of Neurology, Kazan State Medical Academy—Branch Campus of the Federal State Budgetary Educational Institution of Father Professional Education, Russian Medical Academy of Continuous Professional Education, 420012 Kazan, Russia; 3Neurosurgical Department No. 2, Republic Clinical Hospital, 420138 Kazan, Russia; 4Department of Histology, Cytology and Embryology, Kazan State Medical University, 420012 Kazan, Russia

**Keywords:** spinal cord injury, mesenchymal stem cells, extracellular vesicles, systemic administration, local administration, rats

## Abstract

**Simple Summary:**

Spinal cord injury is a serious neurological condition that causes severe disability. The proposed solution of this problem is a cell-free therapy based on using microvesicles derived from stem cells (as an alternative to these cells, to achieve greater safety) intended for neuroregeneration. It was shown that, in the experimental groups with the use of microvesicles, the indicator of motor activity was higher than in the control ones. The best results were found in the group with intravenous administration of microvesicles in comparison with local administration (spinal cord area), where the recovery rate of motor function increased by more than two times compared to the control group.

**Abstract:**

Spinal cord injury (SCI) is a serious neurological condition that causes severe disability. One of the approaches to overcoming the complications of SCI is stem cell-derived extracellular vesicle (EV) therapy. In this research, we performed a comparative evaluation of rat spinal cord post-traumatic regeneration efficacy using different methods of mesenchymal stem cell-derived EV transplantation (local vs. systemic) followed by evaluation of their minimal therapeutic dose. The results suggested that MSC-EV therapy could improve locomotor activity over 60 days after the SCI, showing a dose-dependent effect on the recovery of spinal cord motor pathways. We also established the possibility of maintaining a population of mature oligodendrocytes by MSC-EVs. It was observed that in the spinal cord injury area, intravenous transplantation of MSC-EVs showed more pronounced therapeutic effects compared to the treatment of fibrin matrix-encapsulated MSC-EVs.

## 1. Introduction

Spinal cord injury (SCI) is a severe nerve tissue injury resulting from a mechanical trauma, the incidence and prevalence of which is increasing worldwide every year [1,2,3]. SCI results in a persistent motor and sensory impairment, leading to a significant reduction in the quality of life of patients and, in the worst case, to their death [4]. Standard SCI therapy protocols include stages of neurosurgical care, symptomatic pharmacotherapy and rehabilitation approaches [5]. Due to their low effectiveness, finding new SCI therapy strategies is of paramount importance for both patients and clinicians. 

For many years, due to their high safety and potential neuroprotective properties, mesenchymal stem cells (MSCs) have been used mainly as an experimental therapy to prevent the complications of SCI [6,7]. Numerous studies have shown that MSCs exert therapeutic effects through the secretion of multiple trophic factors and a paracrine influence on the microenvironment, in contrast to the previously considered differentiation and replacement of dead cells. Extracellular vesicles (EVs), containing various bioactive biomolecules, such as DNA, various post-transcriptional regulators, proteins and lipids, play an important role in the implementation of the MSCs’ mechanism of action [8,9,10]. EVs are involved in intercellular communication and are crucial to modulating biological functions within recipient cells, tissue repair and immunomodulation [11,12,13,14,15,16].

The possible positive effect of EVs on neurons and glial cells has already been shown in modeling various neurological diseases. Haney et al. (2015) showed that exogenous EVs are mainly localized with neurons and microglia, and their transplantation has an anti-inflammatory effect, which is expressed as a decrease in microgliosis [17]. With the addition of bone marrow-derived EVs into a primary neuronal culture of APP/PS1 mice (a model of Alzheimer’s disease), the concentration of NO synthase was significantly reduced [18]. In vivo EV transplantation improved cognitive behavior and prevented the impairment of synaptic transmission in the hippocampus in APP/PS1 mice. 

When using EVs obtained from MSCs in a stroke model, a decrease in postischemic motor coordination disorders was noted; exogenous EVs caused long-term neuroprotection, involved in the stimulation of angiogenesis and neurogenesis [19]. Otero-Ortega et al. (2017), in the same model, showed that the intravenous administration of EVs improves functional recovery, axon growth and myelin formation [20]. Moreover, the ability of EVs obtained from bone marrow, adipose tissue, cord blood and dental pulp to induce neuronal outgrowth in an in vitro model of brain injury was shown [21]. In addition, MSC-derived EVs have been shown to be able to modulate microglia towards the neuroprotective M2 phenotype and reduce neuroinflammation [22]. Another study [23] investigated the Ca^2+^ ion-associated intracellular mechanism by examining the neuroprotective effects of MSC-EVs when administered intranasally in two in vivo models of acute brain injury. The authors showed that by activating the PI3K signaling pathway and preventing calcium overload, MSC-EVs protected astrocytes and neurons from cell death under ischemia-like conditions. It can be suggested that MSC-EVs cause the generation of pro-survival signals in astrocytes, which are transmitted to other cells, including neurons.

In the case of SCI, it was shown that EVs are taken up into astrocytes, and the effect of PTEN, which negatively regulates cell division, decreases astrocyte proliferation. Namely, it contributes to the mitigation of SCI by lowering glial scar formation via EV-mediated neuron-astrocyte signaling [24]. Han et al. (2022) demonstrated that EVs from MSCs were able to upregulate the receptor-regulated Smad expression by carrying TGF-β, and Smad 6 knockdown in neural stem cells partially weakened the bone marrow MSC-EV-induced effect on neural differentiation [25]. We can conclude that EVs participate in an extremely wide range of cell activities, play a critical role in cell communication involving neurons and are considered potential therapies and biomarkers for SCI. miRNAs are the most abundant nucleic acids transported by EVs and are effluent cytokines, and they may represent important messengers of EVs and important factors in SCI treatment [26].

Thus, experimental studies to assess the application of EVs in regenerative therapy to overcome the complications of SCI continue to be relevant. In the present study, we performed a comparative evaluation of the efficacy of the rat’s spinal cord posttraumatic regeneration under different MSC-EV introduction conditions (intravenous injection or application to the injury area) with the assessment of their minimal therapeutic dose. 

## 2. Materials and Methods

The animal study was reviewed and approved by the local ethics committee of Kazan Federal University (No. 2, 5 May 2015). The study was designed to minimize the number of animals and the severity of the procedures.

### 2.1. Isolation and Cultivation of Mesenchymal Stem Cells 

The MSCs were obtained from the adipose tissue of female Wistar rats (250–300 g; Pushchino Laboratory, Pushchino, Russia). Surgical manipulations on rats were performed after their anesthesia using isoflurane (1.3%, Laboratories Karizoo, Barcelona, Spain) and zoletil (20 mg/kg, Virbac Sante Animale, Carros, France). Fatty tissue was collected under aseptic operating room conditions in a sterile container with 0.9% NaCl. The adipose tissue was then homogenized, washed by centrifugation in 0.9% NaCl and shredded for 10 min at 1500 rpm. It was then incubated in 0.5% collagenase solution of the crab pancreas (Biolot, Saint Petersburg, Russia) at 37 °C for 1 h, with constant shaking at 180 rpm (on a rocking platform). The homogenate was centrifuged at 1400 rpm for 5 min and the enzyme solution was decanted. The cell pellet was suspended in Dulbecco phosphate salt buffer (DPBS, PanEco, Moscow, Russia) and centrifuged at 1400 rpm for 5 min to remove residual enzymes. The resulting cells were cultured in DMEM medium with 10% fetal bovine serum (FBS), 2 mM L-glutamine, 100 μg/mL streptomycin and 100 U/mL penicillin (all obtained from PanEco, Moscow, Russia). For all cultured cells, the medium was replaced every 3 days. At passage 3, cells were used to obtain EVs.

### 2.2. Isolation of Extracellular Vesicles 

Cells were cultured to a monolayer density of 90–95%. The nutrient medium was then removed, and cells were washed in DPBS and transferred to suspension by treatment with 0.25% trypsin solution. Trypsin inactivation was performed by adding DMEM medium containing 10% FBS. The cells were then centrifuged in falcons for 5 min at 1400 rpm. Cells were washed in 0.9% NaCl to remove serum residues. Cells were then incubated in serum-free DMEM medium containing 10 μg/mL cytochalasin B (Sigma-Aldrich, Burlington, MA, USA) for 30 min (37 °C, 5% CO_2_). After incubation, the cell suspension was actively stirred in a vortex for 60 s and precipitated by centrifugation (500 rpm for 10 min). The resulting supernatant was subjected to two subsequent centrifugation steps, 700 rpm for 10 min and 12,000 rpm for 15 min. The resulting precipitate containing EVs was resuspended in 0.9% NaCl. 

### 2.3. Ultrastructural Assessment of Extracellular Vesicles 

To assess the ultrastructure of the obtained EVs, we fixed it in 2.5% glutaraldehyde. After 12 h from the start of fixation, the samples were placed in a 1% OsO_4_ solution in phosphate buffer with the addition of sucrose, dehydrated and embedded in Epon-812 (Fluka, Charlotte, NC, USA). Ultrathin sections of 0.1 µm thickness were prepared on a Leica UC7 ultramicrotome (Leica, Wetzlar, Germany) and mounted on copper grids (Sigma). Sections were counterstained with uranyl acetate and lead citrate and then examined using a Hitachi 7700 transmission electron microscope (Hitachi, Tokyo, Japan).

### 2.4. Flow Cytometry

The immune phenotype of EVs was analyzed using flow cytometry (BD FACS Aria III, BD Bioscience, East Rutherford, NJ, USA) by immunostaining with the CD49e-PE (Sony, Tokyo, Japan), CD63-FITC (Biolegend, San Diego, CA, USA), Sca1-APC/Cy7 (BioLegend), CD45-PE/Cy7 (BioLegend), CD9-APC (Biolegend) and CD44-APC/Cy7 (BioLegend) antibodies.

### 2.5. Spinal Cord Injury and MSC-EV Therapy 

Experiments were carried out on adult female Wistar rats (weight 250–300 g; Pushchino laboratory, Russia). The animals were kept in transparent plastic cages (12 h daily cycle) with food and water available ad libitum. Surgical manipulations on rats were performed under general anesthesia using isoflurane (1.3%) and zoletil (20 mg/kg). After laminectomy, all animals underwent dosed spinal cord injury (SCI) of moderate severity (2.5 m/s) at the Th8 level using an Impact One Stereotaxic Impactor (Leica) instrument. 

The animals were divided into four experimental groups, in which MSC-EVs were transplanted as follows: animals in experimental groups 1 and 2 received 5 and 10 μg of MSC-EVs encapsulated in 18 μL of fibrin matrix (FM) (Baxter, Deerfield, IL, USA)—SCI FM+EVs5 and SCI FM+EVs10, respectively; animals in experimental groups 3 and 4 had 10 and 50 μg of MSC-EVs injected intravenously (into tail vein) in 500 μL of 0.9% NaCl—SCI EVs10 and SCI EVs50, respectively, 30 min after the injury. Animals in control group 1 were not treated with MSC-EVs (SCI), and animals in control group 2 were injected with FM not containing MSC-EVs (SCI FM) immediately after lesions. Intact rats of appropriate sex and age were examined as a reference control. After surgery, all experimental rats were injected daily with an intramuscular dosage of gentamicin (25 mg/kg) for 7 days. Bladders of the experimental rats were manually emptied twice a day until spontaneous urination appeared. 

### 2.6. BBB Locomotor Rating Scale 

The BBB behavioral scale [27] was used to assess the effectiveness of motor functional recovery. To evaluate the difference in the functional recovery of experimental animals, behavioral assessment in all groups was performed before the SCI, on day 7 after the injury and one day later. Locomotor activity was evaluated simultaneously by two observers who were unaware of the animals’ affiliation with any experimental group. The final results were obtained by averaging the two scores awarded by the observers.

### 2.7. Recording of Electrophysiological Parameters 

Neuromotor functions were assessed using stimulated electromyography. Electrophysiological parameters were recorded in intact rats before the surgery and 30 and 60 days after the SCI. The M- and H-waves of the tibialis gastrocnemius were recorded in response to stimulation of the sciatic nerve in the studied animals. Monopolar needle electrodes were used as active and reference electrodes. The active electrode was inserted in the middle of the muscular abdomen, the reference electrode in the area of the Achilles tendon–muscular junction. Electrical stimulation of the sciatic nerve was performed with a square-wave single stimulus and a pulse width of 0.2 ms. A pair of monopolar needle electrodes injected subcutaneously in the area of sciatic nerve outlet from the pelvis was used for stimulation. 

After recording the M- and H-waves, transcranial electrical stimulation of the cerebral cortex was performed using subcutaneous needle electrodes to record motor-evoked potentials (MEPs). The recording electrodes remained in the gastrocnemius muscle. Stimulation of the cortex was performed using needle electrodes inserted under the scalp skin to reach the skull bone. The cathode was placed along the midline 0.5 cm beyond the interorbital line, the anode along the midline near the occipital bone. The stimulus used had a duration of 0.04 to 0.1 ms and intensity of 20 to 500 V. 

Somatosensory evoked potentials (SEPs) were recorded by monopolar needle electrodes inserted subcutaneously to evaluate conduction across the posterior columns of the spinal cord. An active electrode was inserted over the upper lumbar vertebrae and a reference electrode was inserted over the middle thoracic vertebrae for registration from the lumbar level. An active electrode was inserted in the middle, approximately 0.5 cm caudally from the interorbital line, and a reference electrode in the middle near the occipital bone for registration from the cortex. Electrical stimulation of the tail was performed using ring electrodes. Rectangular electric stimuli of 0.2 ms with a frequency of 3 Hz were used. The stimulus intensity was chosen based on the tail movements (smallest stimulus producing tail movements was used). An average of three repetitions and close in latency responses were analyzed and compared. 

### 2.8. Morphological Analysis 

At 60 days after the SCI, the animals were anesthetized and subjected to intracardiac perfusion with 4% PFA (4 °C). The spinal cord isolated from the spinal column was additionally fixed in 4% PFA at room temperature overnight, followed by incubation in 30% sucrose solution. Samples were placed in a tissue-freezing medium (Tissue-Tek O.C.T. Compound, Sakura, Tokyo, Japan), and then 20-μm-thick cross-sections of the spinal cord were obtained using a Microm HM 560 cryostat (Thermo Fisher Scientific, Waltham, MA, USA). The spinal cord sections were stained with azure and eosin (MiniMed, Bryansk, Russia) at 5 mm from the epicenter of injury/Th8 in the rostral and caudal directions. Stained sections were embedded in a vitrogel and examined using an APERIOCS2 light scanning microscope (Leica, Wetzlar, Germany) according to the previously described technique [28]. During the study, the area of preserved nerve tissue and the total area of pathological cavities were estimated by counting the cavities with an area of at least 1500 µm2. The Aperio ImageScope 12.4 software (Leica) was used for morphometric analysis. 

### 2.9. Immunofluorescence Analysis 

For immunofluorescence analysis, sections were washed with a PBS solution containing Triton X-100. The blocking solution consisted of a PBS solution containing 5% donkey serum albumin. The primary antibodies used were mouse anti-NG2 (Santa Cruz, 1:100) and rabbit anti-Olig2 (Santa Cruz, 1:75). These antibodies were diluted in a blocking solution and incubated at 4 °C overnight. Donkey anti-mouse Alexa Fluor 555 (Abcam, 1:200) and donkey anti-rabbit Alexa Fluor 647 (Abcam, 1:200) were used as secondary antibodies, in which sections were incubated at room temperature for 2 h in the dark. Cell nuclei were stained with 4′6-diamidino-2-phenylindole (DAPI, Sigma, 0.5 µg/mL). Sections were mounted in Mounting Medium for IHC (Abcam, Cambridge, UK) and the following areas of the spinal cord were examined using a LSM 700 confocal microscope (Carl Zeiss, Oberkochen, Germany): ventral horns (VH), ventral (VF) and lateral funiculus (LF). 

### 2.10. RNA Isolation, cDNA Synthesis and Real-Time PCR 

Total RNA was isolated from the rat spinal cord (1 mm segments at a distance of 5 mm caudally from the epicenter of injury) using modified phenol/chloroform extraction according to the manufacturer’s instructions. The quality and quantity of RNA was assessed with NanoDrop (Thermo Fisher Scientific, Waltham, MA, USA). Single-stranded cDNA was synthesized using 100 ng of total RNA, 100 pmol of random hexamer primers, 100 units of MMLV reverse transcriptase (Evrogen, Moscow, Russia) and 5 units of RNAse inhibitor, according to the manufacturer’s protocol (25 °C-10 min, 42 °C-60 min, termination of transcription, 70 °C-10 min). Quantification of 18s, Olig2, Ng2, Mbp, Gap43, Psd95, s100b, S100a10, NeuN and C3 gene mRNA was performed using the CFX384 Touch Real-Time PCR System (Bio-Rad, Hercules, CA, USA). Amplification conditions were as follows: pre-denaturation 95 °C-3 min; 39 cycles: 95 °C-10 s, 55 °C-30 s, 72 °C-30 s, including plate reading; final elongation 72 °C-2 min. We analyzed 100 ng of cDNA to assess the expression of target genes, using 200 nM of each primer and 100 nM of 5x qPCRmix-HS SYBR (Evrogen, Moscow, Russia). mRNA expression was normalized to 18s rRNA. The mRNA level in the intact spinal cord was counted as one. The process was carried out in three replicates. 

### 2.11. Statistical Analysis 

We used R version 3.5.3 (R Foundation for Statistical Computing, Vienna, Austria) for all steps of analysis concerning the morphometric and behavioral data. To evaluate the effect of treatments on gray and white matter preservation after the SCI, a linear mixed-effect model was implemented, with the individual animal as a random variable. Least square means marginalized over the distance from the injury locus in both rostral and caudal directions were applied to estimate the proportion of preserved gray matter in each group (with 95% t confidence intervals for means). We used a nonparametric estimator and a null hypothesis test proposed by [29] as a method for assessing changes in BBB scores over the 60-day period after the SCI. The nonparametric effect size measure could be interpreted as the estimated probability that a randomly selected observation from the mean distribution was smaller than a randomly selected observation from a specific group at a specific time point. Asymptotic 95% confidence intervals for relative marginal effects were also provided as a reference.

The Origin Pro software was utilized to process the other results obtained. Data were presented as mean values and standard deviation/standard error. A normality test was performed on all study groups. One-way analysis of variance (ANOVA) with Tukey’s test was used for multiple comparisons across all groups. All analyses were presented in a blind manner to the study groups. Significant differences below the value of 0.05 (*p* < 0.05) were applied for all statistics. 

## 3. Results

### 3.1. Assessment of Extracellular Vesicles

Flow cytometry analysis showed that the MSC-EVs have surface receptors similar to that of the parental MSCs and express following the receptors: Sca-1 (52% positive), CD49e (67% positive) and CD44 (55% positive). Transmission electron microscopy analysis showed that EVs are homogeneous in size but various in shape and content. In particular, individual multilamellar bodies are visualized in the composition of EVs (Figure 1). The distribution of the diameter of the obtained EVs showed that EVs with a diameter in the range from 0.1 to 0.5 μm prevailed (more than 60%), and this was consistent with previously obtained data [30].

### 3.2. Assessment of Locomotor Activity 

The results indicated that the intravenous administration of EVs showed the highest success rate in improving the motor function of hindlimbs. In particular, days 32 to 60 of the experiment resulted in consistently higher (*p* < 0.05) locomotor scores in the SCI EVs10 and SCI EVs50 groups as compared to the SCI group, according to the BBB scale (Figure 2A). There was no significant difference in locomotor scores across the groups with the intravenous administration of EVs and other types of administration to the SCI area. At 60 days after the SCI, the mean locomotor scores in the experimental groups were 7.71 ± 1.36 (SCI EVs10), 7.63 ± 1.5 (SCI EVs50), 5.0 ± 1.18 (SCI FM+EVs5) and 4.5 ± 0.87 (SCI FM+EVs10). 

It is worth noting that FM application without EVs also contributed to improved motor recovery, having slightly higher locomotor scores compared to the SCI group. In this regard, there was no significant difference across the experimental groups with the FM application with or without EVs (Figure 2B). However, during the 30- to 50-day period, there were periodic reliable differences in hindlimb locomotor scores in the SCI FM+EVs5 and the SCI FM+EVs10 groups compared to the SCI group.

### 3.3. Electrophysiology Results 

Analysis of M-wave amplitudes showed that, on days 30 and 60, the index in the SCI EVs10 and SCI EVs50 groups, respectively, was significantly higher (*p* < 0.05) when compared to other studied groups (Figure 2C). We also obtained a significant decrease (*p* < 0.05) in the amplitude of the M-wave on day 60 compared to day 30 in the SCI EVs10 group and an increase in the SCI EVs50 group. As for the H-wave, we were not always able to evoke it in experimental animals, both on the 30th and 60th day of SCI. Therefore, on day 30, the H-wave was recorded less frequently in the SCI and SCI FM groups compared to the intact controls. There were no intergroup differences observed in the ratio of H-wave amplitude to M-wave amplitude at day 30 and 60 (Appendix A). 

The study of the central motor pathways by transcranial electrical stimulation in all groups on days 30 and 60 revealed a significant decrease in the frequency of MEP registration of the calf muscle during motor cortex stimulation in all groups compared to the intact control. Furthermore, on day 60 of the SCI, the frequency of MEP registration was significantly lower (*p* < 0.05) than on day 30 in all experimental groups. 

There was no difference in the ratio of the M-wave amplitude to the MEP response amplitude (M/MEP) between the experimental groups on day 30 of the SCI. However, the study revealed a lower (*p* < 0.05) ratio of M/MEP amplitudes at day 60 in the SCI EVs50 and SCI FM+EVs5 groups compared to the SCI and SCI FM groups. 

In all intact control animals, SSEP peaks from electrodes located in the lumbar region and under the skin of the scalp were recorded. On the 30th day after the SCI, lumbar peaks continued to be recorded in all animals, without differences in amplitude across the groups, reflecting the preservation of the peripheral region somatosensory pathway of the peripheral region. However, at day 60, there was a decrease (*p* < 0.05) in the amplitude of the lumbar peak in the SCI group compared to the SCI FM and SCI FM+EVs5 groups (Figure 2D). During the same period, the amplitude of the lumbar peak was lower (*p* < 0.05) in the SCI EVs10 group compared to the SCI FM, SCI FM+EVs5 and SCI FM+EVs10 groups. In all studied groups, there was a significant decrease in the frequency of cortical SSEP peaks compared to the intact control, without intergroup differences, at days 30 and 60. 

### 3.4. Morphometrical Assessment

We analyzed the total area of pathological cavities and preserved spinal cord tissue in the experimental groups within 5 mm in the rostral and caudal direction from the epicenter of injury. The area of pathological cavities in the SCI EVs10 group at the injury epicenter and 1 mm rostral from it was significantly lower (*p* < 0.05) than in the SCI group (Figure 3A,E). Similarly, in the SCI FM+EVs5 group, it was significantly lower at 3 mm rostral compared to the SCI FM group (Figure 3B). At the same time, there was no significant difference in the total area of pathological cavities between the groups, regardless of the EV introduction method to the SCI area.

The preserved tissue area assessment showed that, at 5 mm caudally and rostrally to the epicenter of the injury, the index was higher (*p* < 0.05) in the SCI EVs10 group than in the SCI and SCI EVs50 groups (Figure 3C). A positive dynamic was also observed in the SCI EVs10 group at 2 to 4 mm caudally, where the preserved tissue area was significantly greater compared to the SCI group. At the injury epicenter, as well as 2 to 4 mm caudally from it, the preserved tissue area in the SCI FM+EVs10 group was higher (*p* < 0.05) than in the SCI FM+EVs5 group, where fewer EVs were introduced (Figure 3D,E). At the same time, at a distance of 1 mm caudally and 1–5 mm rostrally, the preserved tissue area in the SCI FM+EVs10 group was significantly higher compared to the SCI FM control group.

In addition to the absolute values obtained for the above values, we also evaluated the effect of treatments on the preservation of gray and white matter after the SCI linear mixed-effect model, with individual animals as a random variable. The results of this analysis showed that the percentage of preserved tissue in the SCI FM+EVs10 and SCI EVs10 groups was significantly higher (*p* < 0.05) when compared to the SCI and SCI FM control rat groups (Figure 3F).

### 3.5. Immunofluorescence Results

We performed a quantitative analysis of precursor cells of oligodendrocytes (Olig2+/NG2+) and mature oligodendrocytes (Olig2+/NG2−) in the white (VF and LF) and gray (VH) matter of the spinal cord in experimental rats (Figure 4A–D). There was a reduction in Olig2+/NG2− cells in the spinal cord regions observed, caused by the SCI. However, both methods of EV transplantation contributed to the maintenance of the spinal cord cell population, and a significant difference was found in the experimental groups compared to the control SCI FM in the spinal cord white matter (Figure 4E–G). It was noted that after the maximum dosage of EVs was injected (SCI EVs50), the amount of Olig2+/NG2− cells in the VF and LF zones was greater (*p* < 0. 05) than in the other experimental groups at 60 days after the SCI (Figure 4A,B) and was close to that in the intact control in VF (SCI EVs50: 9.07 ± 1.00 (VF); 9.27 ± 1.72 (LF) vs. intact: 12.93 ± 1.27 (VF); 20.06 ± 2.25 (LF)).

Prominent changes in the population of Olig2+/NG2+ cells were also found in the white matter. In the VF zone, there was an increase in the number of Olig2+/NG2+ cells in the group that received an intravenous injection of the maximum EV dosage (SCI EVs50), when compared to the other groups, with a significant difference when compared to the intact, SCI FM+EVs10 and SCI EVs10 groups (SCI EVs50: 5.07 ± 0.71 vs. Intact: 2.00 ± 0.63; SCI FM+EVs10: 3.67 ± 1.07 and SCI EVs10: 2.92 ± 0.54) (Figure 4C). In the LF zone, the smallest number of LF cells was found in the SCI FM+EVs5 group (Figure 4D,L). In other experimental groups, at 60 days after the SCI, the number of Olig2+/NG2+ cells was comparable to that in the intact control group.

### 3.6. Analysis of mRNA Expression

The expression of *Olig2, Ng2, Mbp, Gap43, Psd95, Neun, s100b, S100a10* and *C3* mRNA genes was assessed in the spinal cord tissue caudally to the epicenter of injury on day 60 after the SCI in the experimental groups and the intact controls. Expression of the mRNA of the *Olig2* gene was highest in the SCI EVs10 group and showed a significant difference between SCI FM and SCI EVs50 (SCI EVs10: 40.78 ± 20.39 vs. SCI FM: 18.23 ± 8.15; SCI EVs50: 18.62 ± 10.75). However, it should be noted that the mRNA expression of the *Olig2* gene was lower (*p* < 0.05) in the SCI group (without FM application and/or EV therapy) compared to SCI FM, SCI FM+EVs10, SCI EVs10 and SCI EVs50 (Figure 4N). Despite the low values of the relative amount of mRNA of *Olig2* in the group that received an intravenous injection of the EVs at the maximum dosage (SCI EVs50), the mRNA expression of the *Ng2* gene was highest in this group, with a significant difference from the SCI, SCI FM and SCI FM+EVs5 groups (Figure 4O). At the same time, *Ng2* mRNA expression was lowest in the SCI group (without FM application and/or EV therapy).

Analysis of *Mbp* and *Gap43* mRNA expression showed a tendency for these values to increase as the dosage of transplanted EVs increased as well (Figure 5A,B). The lowest relative amounts of *Olig2* and *GAP43* mRNA were observed in the SCI group, with a significant difference when compared to other experimental groups. Both groups with the intravenous injection of EVs also showed higher (*p* < 0.05) values of *Mbp* mRNA expression compared to the SCI FM and SCI FM+EVs5 groups.

*Psd95* mRNA expression in the SCI group (without FM application and/or EV therapy) was significantly lower compared to other experimental groups (Figure 5C). At the same time, the maximum value of the relative amount of *Psd95* mRNA was observed in the SCI EVs10 group, where this index was higher (*p* < 0.05) compared to the SCI FM and SCI FM+EVs5 groups.

Interesting dynamics were observed in the SCI FM and SCI FM+EVs5 groups, where the mRNA expression of the Neun, S100a10 and C3 genes was significantly lower compared to other experimental groups, but no significant difference between the two groups was found (Figure 5D–F). At the same time, the relative amount of S100b mRNA was significantly higher in the intravenous EV groups compared to the SCI, SCI FM and SCI FM+EVs5 groups (Figure 5G).

## 4. Discussion

Early studies showed that one of the potential mechanisms of transplanted MSCs’ therapeutic effect might be the secretion of a wide spectrum of active biomolecules, rather than the differentiation and replacement of dead cells in damaged tissue [28,31]. The major components of paracrine factors released by MSCs are EVs [32,33]. RNA and proteins packaged in EVs are stable, which increases their potential application in clinical therapy, including the possibility to pass through the blood–brain barrier. In this study, we first analyzed the effect of MSC-EVs on the structure and function of the injured rat spinal cord considering different doses and methods of their transplantation (local vs. systemic).

Ruppert et al. (2018) demonstrated that the intravenous administration of MSC-EVs leads to the significant recovery of locomotor function and improved sensitivity 14 days after SCI [34]. Our study confirmed these previous findings and demonstrated the possibility of improving locomotor activity parameters up to 60 days after the SCI. The intravenous variant of MSC-EV transplantation was found more beneficial for locomotor recovery compared to MSC-EVs together with FM when applied to the injured area.

Currently, there are various matrices (i.e., ECM matrix, collagen type I matrix) that are commercially available for nerve gap repair. Previously, it was shown that the application of FM (Baxter) during surgical treatment in patients with SCI was effective in reducing postoperative complications and in shortening the duration of the hospital stay, with a consequent saving in costs [35]. According to the manufacturer’s instructions, the clot of fibrin glue formed during preparation is completely absorbed during wound healing, which ensures the release of the contents contained in the matrix. It was also shown that ChABC, enclosed in the abovementioned FM, is released into the tissue of the spinal cord, and nearly six-times more bioactive ChABC was detected in the spinal cord 3 weeks after injury when the fibrin delivery system was used vs. an intraspinal injection of ChABC [36]. However, it cannot be ruled out that the fibrin gel employed may not be optimal for nerve gaps since it is a hemostatic agent and other alternatives could be more suitable; the release kinetics were not studied and should be assessed in future studies.

Summarizing the results of the electrophysiological analysis, the MSC-EVs had a positive effect on the motor parameters of the nervous system, both central and peripheral. The effect was dose-dependent: the higher doses of MSC-EVs were superior to the lower doses of MSC-EVs in maintaining or improving spinal cord motor function after injury. The MSC-EVs also showed a small positive effect on the sensory parts of the nervous system distal to the injury site.

The morphometric analysis revealed no dose-dependent effect on tissue preservation. The highest percentage of preserved tissue was found in the SCI FM+EVs10 and SCI EVs10 groups. A study by Romanelli et al. (2019) investigated the long-term effects of MSC-EVs transplanted intravenously or intraparenchymally in an SCI rat contusion model [37]. The work showed that, up to 56 days post-injury, both the experimental and control groups continued to lose preserved nerve tissue. By the last day of the experiment, the intraparenchymal MSC-EV group showed significantly more preserved tissue compared to the vehicle-treated group.

We first observed an increase in the number of mature Olig2+ cells during MSC-EV therapy in the white matter of the spinal cord, which may indicate the stimulation of nerve fiber myelination. Our results indicating the maintenance of a mature oligodendrocyte population by MSC-EVs were also confirmed by PCR-RT data. Thus, we found that the mRNA expression of the *Mbp* gene encoding myelin-forming protein synthesis was increasing with the growing dosage of transplanted MSC-EVs. Go et al. (2021) quantified the oligodendrocyte progenitor cell density (NG2+ OPC) in a cortical injury model in aged rhesus monkeys and found no significant difference between the MSC-EV (4×10^11^ particles/kg) intravenously injected group and the control group [38]. In our work, we also did not find a significant increase in the number of Olig2+/NG2+ cells in the studied areas between the MSC-EV transplantation groups and the SCI FM control group. At the same time, *Ng2* mRNA expression tended to increase significantly with the MSC-EV transplantation in a dose-dependent manner.

Evaluation of astrocyte reactivity in the context of EV therapy in SCI is always a matter of interest among researchers. Therefore, Noori et al. (2021) performed an intrathecal injection of 2 or 3 μg of MSC-EVs in a compression SCI rat model, resulting in decreased astrocytic activity, stimulation of neural progenitor cells and the protection of more cells from death in the acute period of SCI (7 dpi) [39]. Other research groups working with SCI rat models demonstrated that the intravenous injection of MSC-EVs led to the suppression of neurotoxic reactive astrocyte A1 activation compared to the control group [40,41]. In our study, we did not quantify different astrocyte cell populations, but we did determine the mRNA expression levels of the C3 and S100a10 genes encoding the synthesis of the corresponding proteins specific to A1/A2 astrocytes, respectively. Interestingly, we detected a similar trend indicating a significant decrease in the expression of these genes in the SCI FM and SCI FM+EVs5 groups. In other experimental groups, the level of C3 and S100a10 gene mRNA expression was higher and had no significant difference from the SCI control group after the transplantation of high doses of MSC-EVs. It was shown that the addition of MSC-EVs to the neuroglia culture can influence/modulate intracellular calcium fluxes in astrocytes but not in neurons [23]. There are many data showing the participation of calcium ions in hereditary and acquired neuropathogenesis. In most cases, this is attributed to an undesirable increase in intracellular calcium, which is called cytosolic calcium overload. This fact offers a basis to consider the mechanisms of calcium ion transport into the cell and related intracellular processes as a separate aspect of the development of a neurotherapeutic strategy. An increase in the relative expression of the mRNA of astrocyte genes may indirectly indicate an increase in the number of A1/A2 astrocytes. Considering the positive effect of EV supplementation on astrocytes in the context of their protective functions against calcium overload, we can relate these observations to each other and suggest that an increase in astrocyte mRNA expression in groups with EV intravenous administration correlates with a positive effect on neuroregeneration during EV therapy.

## 5. Conclusions

Thus, our data demonstrated the higher beneficial therapeutic effect of intravenously transplanted MSC-EVs after SCI according to the assessment of locomotor activity, morphometrical assessment and the number of myelinating mature Olig2+ cells. However, we did not always detect dose-dependent effects, which may be due to (1) the variability of the EVs’ secretome profile and (2) the inverse effect of a dose rate. In conclusion, it should be assumed that additional research involving large animals (pigs, monkeys) will be able to provide more answers to the existing questions. The results obtained will allow us to implement autologous variants of MSC-EV transplantation and circumvent some of the problems related to structural and physiological characteristics between humans and small animals.

## Figures and Tables

**Figure 1 biology-11-01853-f001:**
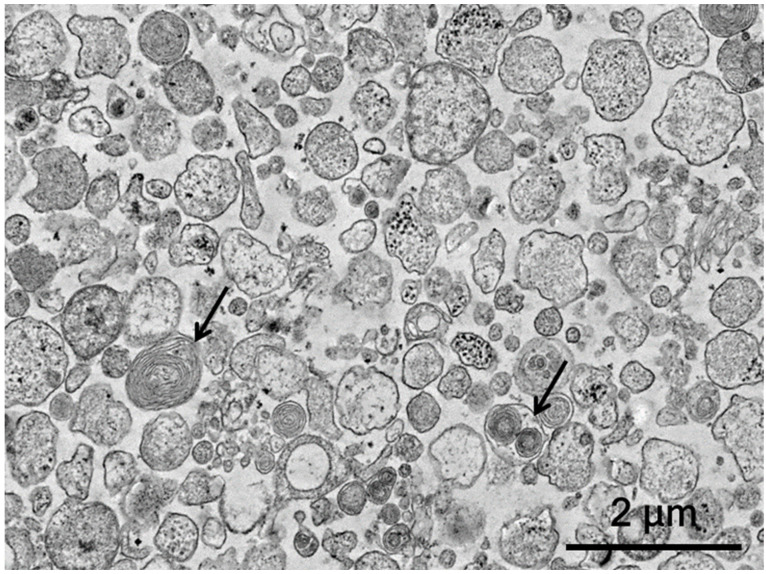
Analysis of the morphology of the MSC-EVs. Transmission electron microscopy was used to characterize the obtained EVs; arrows—multilamellar bodies.

**Figure 2 biology-11-01853-f002:**
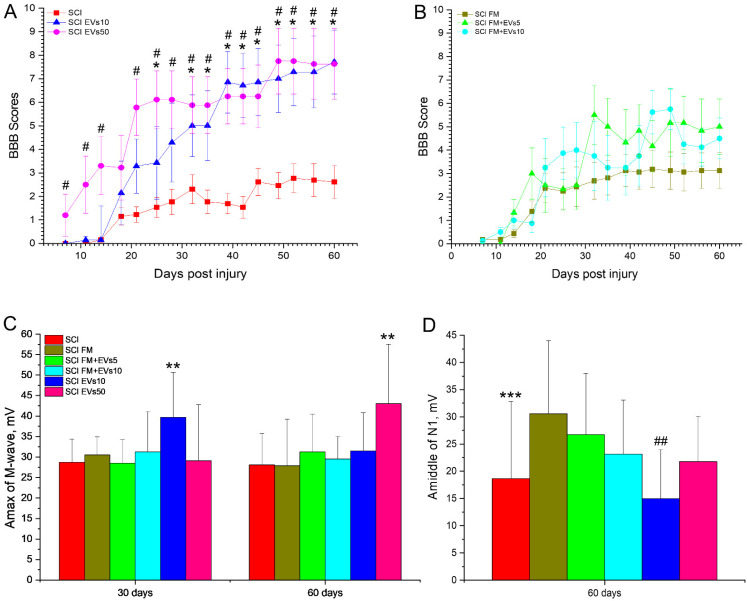
(**A**,**B**) Results of functional tests on the “BBB” scale in scores (Y axis), from 7 to 60 days after SCI in the experimental groups (X axis). # *p* < 0.05—SCI compared to SCI EVs50; * *p* < 0.05—SCI compared to SCI EVs10. (**C**) Results of the evaluation of the amplitude of M-waves (Y axis) at 30 and 60 days after SCI in the experimental groups (X axis). ** *p* < 0.05 compared to all groups. (**D**) The middle of lumbar N1 (Y axis) after SCI in the experimental groups (X axis) at 60 days. *** *p* < 0.05—SCI compared to SCI FM and SCI FM+EVs5; ## *p* < 0.05—SCI EVs10 compared to SCI FM, SCI FM+EVs5 and SCI FM+EVs10.

**Figure 3 biology-11-01853-f003:**
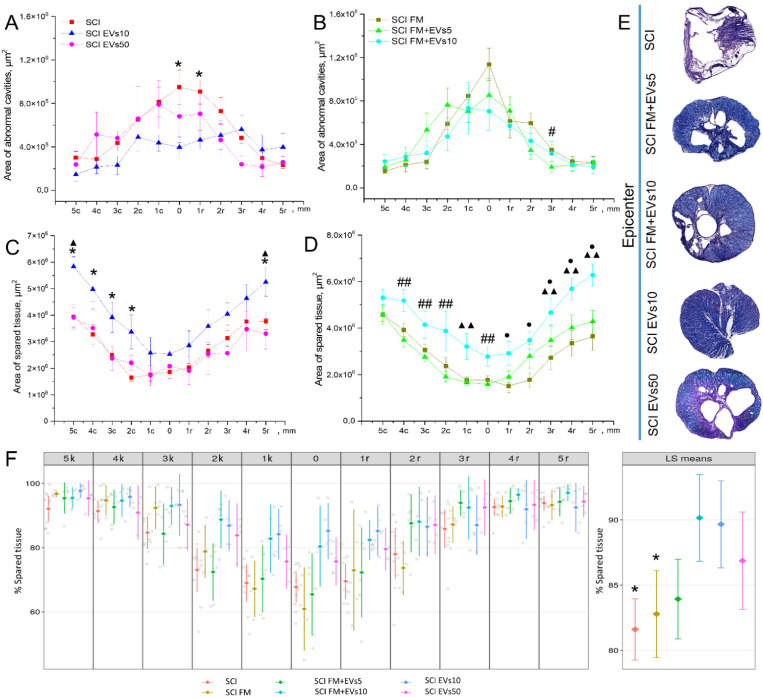
Results of morphometric analysis with evaluation of (**A**,**B**) total area of pathological cavities and (**C**,**D**) area of preserved tissue of the spinal cords of rats of the experimental groups at 5 mm in the rostral and caudal direction (X axis) at 60 days from the SCI epicenter. * *p* < 0.05—SCI compared to SCI EVs10; # *p* < 0.05—SCI FM compared to SCI FM+EVs5; ## *p* < 0.05—SCI FM+EVs5 compared to SCI FM+EVs10; ▲ *p* < 0.05—SCI EVs50 compared to SCI EVs10; ▲▲ *p* < 0.05—SCI FM+EVs10 compared to SCI FM and SCI FM+EVs5; ● *p* < 0.05—SCI FM compared to SCI FM+EVs10. (**E**) Cross-sections of the injured spinal cord at 60 days after SCI in experimental groups at the injury epicenter. Azur–eosin staining. (**F**) Left—Group means (with 95% t CIs) of preserved gray matter after SCI at different distances from the injury locus. Gray dots represent raw data. Right—Group least square means marginalized over the distance from the injury locus (with 95% t confidence intervals). * *p* < 0.05—comparing SCI FM+EVs10 and SCI EVs10.

**Figure 4 biology-11-01853-f004:**
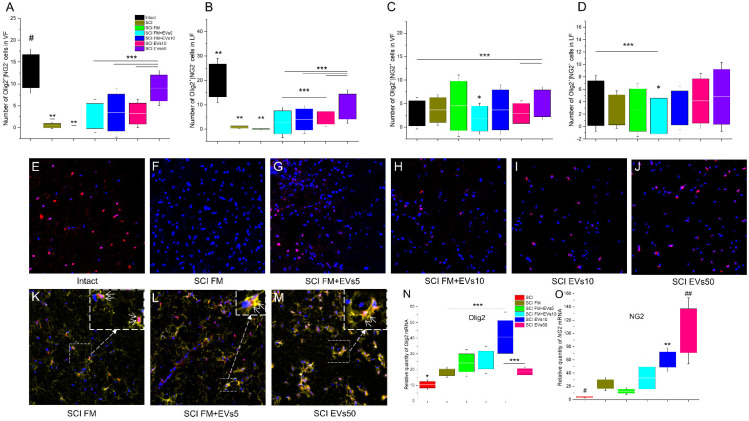
(**A**–**D**) Number of Olig2+/NG2- and Olig2+/NG2+ cells (Y axis) in control and experimental animal groups at 60 days after SCI. * *p* < 0.05—as compared with SCI FM+EVs10, SCI EVs10 and SCI EVs50; ** *p* < 0.05—as compared with all groups; # *p* < 0.05—as compared with SCI FM, SCI FM+EVs5, SCI FM+EVs10 and SCI EVs10; *** *p* < 0.05. (**E**–**J**) Panels illustrate Olig2+/NG2−-cells in intact rats (**E**), Olig2+/NG2−-cells at day 60 after fibrin matrix application (**F**), MSCs-EVs within fibrin matrix (**G**,**H**) and intravenous infusion of MSCs-EVs in saline (**I**,**J**) in LF zone of spinal cord. (**K**–**M**) Panel illustrates Olig2+/NG2+-cells on day 60 after application of fibrin matrix (**K**), MSCs-EVs within fibrin matrix (**L**) and intravenous infusion of MSCs-EVs in saline (**M**). The zones highlighted with dashes correspond to each other, and the arrow points to the enlarged image of the corresponding zone. Arrows indicate Olig2+/NG2+-cells. Scale bar: 20 μm. (**N**,**O**) Relative quantity of *Olig2 and NG2* mRNA expression at 60 days after SCI in experimental groups, calculated in relation to intact spinal cord controls, which were considered as 1. mRNA expression was normalized using 18s rRNA. * *p* < 0.05—as compared with SCI FM, SCI FM+EVs10, SCI EVs10 and SCI EVs50; ** *p* < 0.05—as compared with SCI FM and SCI FM+EVs5; # *p* < 0.05—as compared with SCI FM, SCI FM+EVs5 and SCI EVs10; ## *p* < 0.05—as compared with SCI, SCI FM and SCI FM+EVs5; *** *p* < 0.05.

**Figure 5 biology-11-01853-f005:**
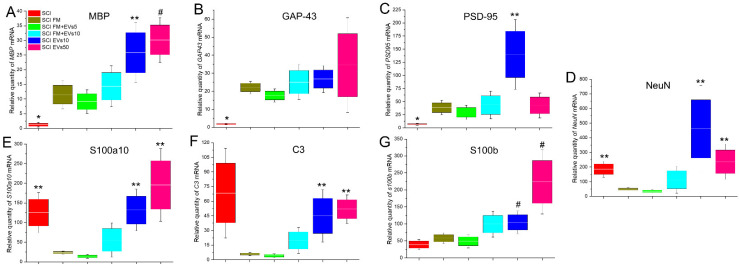
Analysis of mRNA expression in the caudal area of SCI in experimental rats. (**A**–**G**) Relative quantity of expression of *MBP, GAP43, PSD95, NeuN, s100b, S100a10* and *C3* mRNA 60 days after SCI, calculated in relation to intact controls, which were considered 1. mRNA expression was normalized using *18s* rRNA. * *p* < 0.05—as compared with all groups; ** *p* < 0.05—as compared to SCI FM and SCI FM+EVs5; # *p* < 0.05—as compared to SCI, SCI FM and SCI FM+EVs5.

## Data Availability

Not applicable.

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
