# Peer review of "A Comparative Study of Mesenchymal Stem Cell-Derived Extracellular Vesicles’ Local and Systemic Dose-Dependent Administration in Rat Spinal Cord Injury"

_biology, 2022, doi:10.3390/biology11121853_

Round 1

Reviewer 1 Report

The work is interesting and the authors obtained new results. I have a few comments.

1) Was the injection of a conditioned medium that does not contain vesicles 2) I would recommend arranging the figures according to the text

3) The introduction should reflect the effects of vesicles on neurons and astrocytes.

4) The conclusion must strictly reflect the results obtained. References to works of other authors are not appropriate. In general, the conclusion should be clearer based on the results obtained.

5) The role of Ca2+-signaling of astrocytes under the action of vesicles should be discussed. https://pubmed.ncbi.nlm.nih.gov/36147480/

Author Response

The work is interesting and the authors obtained new results. I have a few comments.

Authors: We would like to thank the reviewer for this review and pointing out some errors. We hope that our following responses will satisfy the reviewer.

1) Was the injection of a conditioned medium that does not contain vesicles

Authors: No, we did not inject vesicle-free conditioned medium because we aimed to compare the effectiveness of the transplantation method with the identification of its minimum therapeutic dose.

2) I would recommend arranging the figures according to the text

Authors: Done.

3) The introduction should reflect the effects of vesicles on neurons and astrocytes.

Authors: According to the reviewer’s comment, we have added to Introduction a little review about effects of vesicles on neurons and glial cells and highlighted it with yellow marker.

4) The conclusion must strictly reflect the results obtained. References to works of other authors are not appropriate. In general, the conclusion should be clearer based on the results obtained.

Authors: According to the reviewer’s comment, we have edited conclusion.

5) The role of Ca2+-signaling of astrocytes under the action of vesicles should be discussed. https://pubmed.ncbi.nlm.nih.gov/36147480/

Authors: We have added the role of Ca2+-signaling of astrocytes under the action of vesicles to Introduction and Discussion.

We would like to thank the reviewer for this review and pointing out some errors. We hope that our responses will satisfy the reviewer.

Reviewer 2 Report

The presented study investigates the therapeutic effects of MSC-derived EVs on SCI, comparing effects of dosage and delivery method (i.e. local vs systemic).  The study scope is interesting and relevant to the research field, however some main points need to be addressed before publication.

1)      The main issue with the manuscript is the lack of the assessment for the EVs. EV isolation method is very general (12,000 rpm), where most likely most of vesicles in the range of 100-400 nm have been discarded (these require higher speeds for collection). Besides, there is no analysis on these collected EVs at morphologic and concentration level (NTA), characterization (specific markers by western blot or flow cytometry) nor specification of how the protein quantification was carried out. For a dosage dependent research paper, these analysis must be included.

2)      For the local delivery, there is no explanation on the methods on how the fibrin matrix was prepared and, in case this was purchased and used as provided, there is not protocol description for EVs incorporation nor release kinetics. This is also a main issue for a paper that pursues investigating local delivery with a material. Also, what is the rationale for using fibrin matrix over other options (i.e. ECM matrix, collagen type I matrix) which are commercially available for nerve gap repair?

3)      Figure 1. H-waves are fully described in results but no graph is included in the figure. Same for Figure 1D at day 30.

4)      In line 231 the text reads: “These changes were confirmed by a significant decrease (P<0.05) in the amplitude of the M-wave on day 60 compared to day 30 in the SCI EVs10 group and an increase in the SCI EVs50 group”. Why does this reduction in EVs10 and increase in EVs50 confirm the results on day 30?

5)      Figure 3 A-D does not include a SCI control, whereas this control is included in all figures throughout the study. Could authors offer an explanation? If not this control must be included.

6)      Discussion should be more elaborated. In the current version, the discussion just compares these results to previous studies, while does not investigate any plausible mechanism. Also, the study presents discrepancies between figures (i.e. mRNA levels and expression of markers quantifications, functional electrophysiological readings and  markers expression etc.). This discrepancies can be understood but must be discussed.

Minor comments:

7)      Title: Should be “Dose-dependent administration”

8)      SCI control should be also included in figures 1B, 2B, 2D.

9)      Figures 3 and 4 are very difficult to read. Please consider re-distributing them or making the labels bigger. Also, a title with the name on top of each marker/gene graph would be helpful.

Author Response

The presented study investigates the therapeutic effects of MSC-derived EVs on SCI, comparing effects of dosage and delivery method (i.e. local vs systemic). The study scope is interesting and relevant to the research field, however some main points need to be addressed before publication.

Authors: We would like to thank the reviewer for this review and pointing out some errors. We hope that our following responses will satisfy the reviewer.

1) The main issue with the manuscript is the lack of the assessment for the EVs. EV isolation method is very general (12,000 rpm), where most likely most of vesicles in the range of 100-400 nm have been discarded (these require higher speeds for collection). Besides, there is no analysis on these collected EVs at morphologic and concentration level (NTA), characterization (specific markers by western blot or flow cytometry) nor specification of how the protein quantification was carried out. For a dosage dependent research paper, these analysis must be included.

Authors: To obtain EVs, we used the standard proven method of our laboratory for its isolation from MSCs (Gomzikova et al., 2020, doi: 10.1038/s41598-020-67563-9). Here, by us the cytokine content and surface marker expression of EVs was determined. In addition, the scanning electron microscopy results demonstrate a pure EVs fraction without inclusions of cell debris.

2)      For the local delivery, there is no explanation on the methods on how the fibrin matrix was prepared and, in case this was purchased and used as provided, there is not protocol description for EVs incorporation nor release kinetics. This is also a main issue for a paper that pursues investigating local delivery with a material. Also, what is the rationale for using fibrin matrix over other options (i.e. ECM matrix, collagen type I matrix) which are commercially available for nerve gap repair?

Authors: The commercial kit "Tissucol" contains 4 vials used to obtain a two-component glue. We obtained the first component by mixing fibrinogen with aprotinin, the second component is obtained by mixing thrombin with calcium chloride. We took the components in equal quantities. 9 µl of a solution of fibrinogen in aprotinin was mixed with the EVs at a temperature of +37°C. Immediately before application to the injury site, 9 μl of thrombin solution with calcium chloride was added to 9 μl of fibrinogen in aprotinin solution containing EVs and applied to the injury site with mechanical pipette.

The fibrin matrix used in our study, also is commercially available and includes coagulating proteins, fibrinogen, plasma fibronectin (CIG), factor XIII, plasminogen, aprotinin solution, thrombin 4, thrombin 500, CaCl2 solution and used routinely in clinical practice. (https://www.baxterhealthcare.co.uk/healthcare-professionals/surgical-care/tisseel-surgical-care# ). According to the manufacturer's instructions, the clot of fibrin glue formed during preparation is completely absorbed during wound healing, which ensures the release of the contents contained in the matrix.

3)      Figure 1. H-waves are fully described in results but no graph is included in the figure. Same for Figure 1D at day 30.

Authors: No significant changes in H-waves were found, and therefore we decided not to make the article heavier with these graphs, but at the request of the reviewer we added it to the supplementary materials (Table 1 and Table 2)

4)      In line 231 the text reads: “These changes were confirmed by a significant decrease (P<0.05) in the amplitude of the M-wave on day 60 compared to day 30 in the SCI EVs10 group and an increase in the SCI EVs50 group”. Why does this reduction in EVs10 and increase in EVs50 confirm the results on day 30?

Authors: We regret our mistake as we have formulated the sentence incorrectly. In accordance with the comment of the reviewer, we have corrected the sentence as follows and highlighted it with a yellow marker: «We also obtained a significant decreasing (P<0.05) in the amplitude of the M-wave on day 60 compared to day 30 in the SCI EVs10 group and an increasing in the SCI EVs50 group».

5)      Figure 3 A-D does not include a SCI control, whereas this control is included in all figures throughout the study. Could authors offer an explanation? If not this control must be included.

Authors: According to the reviewer’s comment, we have added SCI control data to the Figure 3 A-D.

6)      Discussion should be more elaborated. In the current version, the discussion just compares these results to previous studies, while does not investigate any plausible mechanism. Also, the study presents discrepancies between figures (i.e. mRNA levels and expression of markers quantifications, functional electrophysiological readings and markers expression etc.). This discrepancies can be understood but must be discussed.

Authors: According to the reviewer’s comment, we have extended Discussion and highlighted new text with yellow marker.

Minor comments:

7)      Title: Should be “Dose-dependent administration”

Authors: Done.

8)      SCI control should be also included in figures 1B, 2B, 2D.

Authors: SCI control was included originally in figures 1B, 2B, 2D. Displaying all experimental groups on one graph would be difficult to visualize and understand the data. In this regard, to facilitate visualization, we divided the graphs into 2 parts. Thus, Figure 1B is part of Figure 1A, Figure 2B is part of Figure 2A, and Figure 2D is part of Figure 2C.

9)      Figures 3 and 4 are very difficult to read. Please consider re-distributing them or making the labels bigger. Also, a title with the name on top of each marker/gene graph would be helpful.

Authors: We have added all figures with the best original quality to the manuscript.

Round 2

Reviewer 1 Report

The authors have done a lot of work on the article. The article can be accepted for publication

Author Response

The authors have done a lot of work on the article. The article can be accepted for publication

Authors: We would like to thank the reviewer for this review and his work.

Reviewer 2 Report

Authors have addressed most of this reviewer comments, however still two main points of vital importance to the paper are not addressed:

1) Authors did not provide an assessment of the EVs employed in this study. Even though the isolation method to obtain these EVs has been assessed in a previous publication, this information is very relevant to the reader and is not available in the manuscript. Therefore, in an optimal case authors must still include a brief assessment of the EVs employed. Alternatively, authors should cite the reference of the EVs isolation method in the correspondent methods section, and include a brief description of the obtained EVs characteristics (size average +/- standard error, positive markers, ratio protein/particles) in the results. 

2) Authors did not offer a clear explanation on the choice of material and release kinetics. Therefore, a paragraph in the discussion section including these limitations (i.e. the fibrin gel employed may not be optimal for nerve gap since is an hemostatic agent and other alternatives could be more suitable, the release kinetics is not studied and should be assessed in future studies) must be included.

Author Response

Authors have addressed most of this reviewer comments, however still two main points of vital importance to the paper are not addressed:

1) Authors did not provide an assessment of the EVs employed in this study. Even though the isolation method to obtain these EVs has been assessed in a previous publication, this information is very relevant to the reader and is not available in the manuscript. Therefore, in an optimal case authors must still include a brief assessment of the EVs employed. Alternatively, authors should cite the reference of the EVs isolation method in the correspondent methods section, and include a brief description of the obtained EVs characteristics (size average +/- standard error, positive markers, ratio protein/particles) in the results.

Authors: We would like to thank the reviewer for this review and pointing out some errors. We hope that our following responses will satisfy the reviewer.

According to the reviewer’s comment we have added our data on transmission electron microscopy, which describes the morphology of the obtained EVs, as well as data on flow cytometry to the appropriate sections (Materials and Methods and Results), and highlighted the new text with a green marker. In addition, we have added a link to previously published study on EVs.

2) Authors did not offer a clear explanation on the choice of material and release kinetics. Therefore, a paragraph in the discussion section including these limitations (i.e. the fibrin gel employed may not be optimal for nerve gap since is an hemostatic agent and other alternatives could be more suitable, the release kinetics is not studied and should be assessed in future studies) must be included.

Authors: According to the reviewer’s comment we have added a paragraph to the Discussion section, where we added information about the materials and matrixes using in current researches and discussed this issue and highlighted it with a green marker.
